# Impact of Omicron Variant Infection on Assessment of Spike-Specific Immune Responses Using the EUROIMMUN Quan-T-Cell SARS-CoV-2 Assay and Roche Elecsys Anti-SARS-CoV-2-S

**DOI:** 10.3390/diagnostics13061024

**Published:** 2023-03-08

**Authors:** Mohamed I. M. Ahmed, Michael Plank, Noemi Castelletti, Paulina Diepers, Tabea M. Eser, Raquel Rubio-Acero, Ivan Noreña, Christina Reinkemeyer, Dorinja Zapf, Michael Hoelscher, Christian Janke, Andreas Wieser, Christof Geldmacher

**Affiliations:** 1Division of Infectious Diseases and Tropical Medicine, University Hospital, LMU Munich, 80802 Munich, Germany; 2German Center for Infection Research (DZIF), Partner Site Munich, 81675 Munich, Germany; 3Fraunhofer Institute for Translational Medicine and Pharmacology ITMP, Immunology, Infection and Pandemic Research, 80799 Munich, Germany; 4Institute of Radiation Medicine, Helmholtz Munich, German Research Centre for Environmental Health, 85764 Neuherberg, Germany; 5Institute for Experimental Immunology, Affiliated to EUROIMMUN Medizinische Labordiagnostika AG, 23560 Lübeck, Germany; 6Max Von Pettenkofer Institute, Faculty of Medicine, LMU Munich, 80336 Munich, Germany

**Keywords:** SARS-CoV-2, spike-specific immune response, omicron, breakthrough infections

## Abstract

The currently prevailing variants of SARS-CoV-2 are subvariants of the Omicron variant. The aim of this study was to analyze the effect of mutations in the Spike protein of Omicron on the results Quan-T-Cell SARS-CoV-2 assays and Roche Elecsys anti-SARS-CoV-2 anti-S1. Omicron infected subjects ((*n* = 37), vaccinated (*n* = 20) and unvaccinated (*n* = 17)) were recruited approximately 3 weeks after a positive PCR test. The Quan-T-Cell SARS-CoV-2 assays (EUROIMMUN) using Wuhan and the Omicron adapted antigen assay and a serological test (Roche Elecsys anti-SARS-CoV-2 anti-S1) were performed. Using the original Wuhan SARS-CoV-2 IGRA TUBE, in 19 of 21 tested Omicron infected subjects, a positive IFNy response was detected, while 2 non-vaccinated but infected subjects did not respond. The Omicron adapted antigen tube resulted in comparable results. In contrast, the serological assay detected a factor 100-fold lower median Spike-specific RBD antibody concentration in non-vaccinated Omicron infected patients (*n* = 12) compared to patients from the pre Omicron era (*n* = 12) at matched time points, and eight individuals remained below the detection threshold for positivity. For vaccinated subjects, the Roche assay detected antibodies in all subjects and showed a 400 times higher median specific antibody concentration compared to non-vaccinated infected subjects in the pre-Omicron era. Our results suggest that Omicron antigen adapted IGRA stimulator tubes did not improve detection of SARS-CoV-2-specific T-cell responses in the Quant-T-Cell-SARS-CoV-2 assay. In non-vaccinated Omicron infected individuals, the Wuhan based Elecsys anti-SARS-CoV-2 anti-S1 serological assay results in many negative results at 3 weeks after diagnosis.

## 1. Introduction

Severe acute respiratory syndrome coronavirus 2 (SARS-CoV-2) is the causative agent of the current COVID-19 pandemic; with more than half a billion infected individuals and more than 6 million deaths. The virus was first detected in December 2019 in Wuhan, China, and rapidly spread across the world. The currently prevailing variants of the virus are subvariants of the Omicron variant (B.1.1.529), which evolved by July 2022 into BA.1, BA2-5 and BQ1.1, that dominate the pandemic. Omicron differs from previous variants of concern in regard to its infectiousness and the more than 30 different mutations within the Spike protein [1]. However, Omicron and its subvariants are substantially less pathogenic compared to the original Wuhan or Delta strains [2].

Current diagnostic techniques involved in the detection of acute SARS-CoV-2 infections utilize nose or throat swabs, followed by direct viral RNA detection using RT-PCR techniques or direct detection of specific SARS-CoV-2 antigens. Antigen tests have proven to be useful; however, they lack in sensitivity, especially with lower viral loads [3]. To prove past infections with SARS-CoV-2, serologic tests, such as the Roche Elecsys anti-SARS-CoV-2 anti-S1 or anti-nucleocapsid, are used. These tests detect antibodies against SARS-CoV-2 proteins, such as Spike or Nucleocapsid, and can be used with good sensitivity and specificity at high throughput [4]. However, such techniques often still use Wuhan wild type virus antigens. Spike-specific T-cell responses against SARS-CoV-2 appear to be less affected by Spike mutations in Omicron variants [5], which is relevant for immunodiagnosis of infection and for characterization of adaptive immunity in convalescent patients, vulnerable populations (e.g., immunologically impaired individuals or the elderly population) and/or vaccinated subjects [6,7]. 

The Quan-T-Cell SARS-CoV-2 assay is a commercially available interferon gamma release assay (IGRA) which quantifies interferon-y (IFNy), which is specifically released by T-cells upon in vitro restimulation with specific peptides of the Spike antigen [3]. SARS-CoV-2-specific T-cells producing IFNy contribute to immune protection from severe disease in humans [8] and are essential for vaccine-induced protection upon SARS-CoV-1 infection in mice and non-human primates (NHP) [9,10,11,12]. The benefits of IGRA assays in general are their relative simplicity in comparison to ELISPOT and intracellular cytokine staining, making them suitable for assessment of T-cell responses even in resource-limited settings.

This Quan-T-Cell SARS-CoV-2 assay was developed by using N-terminal Spike peptides based on the Wuhan strain in antigenic regions. These regions are affected by mutations occurring in the Omicron variant, which may affect assay accuracy [13]. Indeed, reduced test sensitivity for Omicron variant samples has been demonstrated for several rapid-tests or serology assays when compared to Wuhan and delta variant samples [14].

The aim of this study was to determine whether the currently marketed Quan-T-Cell SARS-CoV-2 from EUROIMMUN could be tailored and improved by adapting the antigenic cocktail with Omicron variant peptides. In order to achieve this, we tested the CE-IVD certified “Wuhan” based stimulator tube against an updated version containing antigens based on the Omicron Spike protein and compared the measured IFNy concentration in the supernatant in a head-to-head comparison. Furthermore, we also tested to what degree Omicron infections, breakthrough infections (BTI) and non-breakthrough infections (non BTI), influence the results of the serological Roche Elecsys anti-SARS-CoV-2 anti-S assay in direct comparison to samples from patients infected during earlier phases of the pandemic, when the Wuhan strain still dominated.

## 2. Materials and Methods

### 2.1. Study Population

The study participants were recruited from the KoCo19-Immu cohort (Project number: 20-371), which is a prospective study started in 2020 and conducted in Munich, Germany. On 1 December 2020, the KoCo19-Immu cohort joined the ORCHESTRA (Connecting European Cohorts to Increase Common and Effective Response to SARS-CoV-2 Pandemic) project.

KoCo19-Immu aims to identify and characterize factors that influence the clinical course and further transmission of SARS-CoV-2 infection. The 37 participants of this KoCo19-sub-study were recruited from December 2021 until the end of March 2022. The general inclusion and exclusion criteria for the KoCoImmu study are based on the ones of the KoCo19 study which have been described in detail previously [15]. Additionally, there were specific criteria used for the purpose of the Omicron subgroup. Only outpatient BTI (vaccinated followed by infection, *n* = 20) and non-BTI (first time infected, *n* = 17) were recruited. Potential participants were questioned prior to the visit and only the ones who reported no previous SARS-CoV-2 infection were included. Furthermore, for the purpose of this analysis, only subjects that were confirmed SARS-CoV-2 PCR positive by routine laboratory diagnostics were considered. Subjects were either reported by the health authorities as confirmed Omicron cases or had a confirmation of Omicron infection by the initial PCR test or a high likelihood of an Omicron infection, which was indicated by testing positive for specific mutation markers. No PCR confirmation or detection of a different virus variant were exclusion criteria. Recruitment into the study followed 3 weeks after the SARS-CoV-2 PCR positive diagnosis. Missing necessary samples were classified as exclusion criteria. Samples from acutely SARS-CoV-2 infected subjects recruited during the early phase of the pandemic (May 2020–January 2021) were matched according to the time since diagnosis for serological comparisons with the Omicron cases.

### 2.2. Quan-T-Cell SARS-CoV-2 Interferon Gamma Release Assay

Blood samples for testing in the Quan-T-Cell SARS-CoV-2 Interferon gamma release assay could be obtained and processed from 26 of all subjects with a SARS-CoV-2 infection by the Omicron strain; out of these, 16 were BTI and 10 were non BTI cases. Samples were collected after a median of 21 days (Range: 11–55) and 28 days (Range: 6–37) after diagnosis for BTI and non BTI, respectively. The participants had a median age of 51 (Range: 25–81) in the BTI group and 50 (Range: 24–59) in the non BTI group. The majority of the study subjects were female (77%, 20/26).

Heparin tubes were used to collect 6 mL of fresh whole blood. A volume of 500 µL was then stimulated overnight (16–18 h) at 37 °C and 5% CO_2_ in the SARS-CoV-2 IGRA BLANK (negative control), STIM (positive control using mitogen), and TUBE (antigens based on the SARS-CoV-2 Wuhan Spike protein) tubes (Quan-T-Cell SARS-CoV-2, EUROIMMUN, Ref: ET 2606-3003) and the SARS-CoV-2 IGRA Omicron (antigens based on the SARS-CoV-2 Omicron Spike protein) tube (EUROIMMUN). Following incubation, the cells were centrifuged at 12,000× *g* for 10 min and the plasma collected and frozen at −80 °C for later IFNy analysis using the Quan-T-Cell-ELISA kit (EUROIMMUN, Ref: EQ 6841-9601) on the fully automated EUROIMMUN Analyzer I (EUROIMMUN).

Background subtraction was performed and, thereafter, IFNy concentrations from the two different measurements were classified into three different categories: (i) negative (<0.1 IU/mL), (ii) borderline (0.1–0.2 IU/mL) and (iii) positive (>0.2 IU/mL). These cut-offs were taken from the CE-IVD certified kit for the SARS-CoV-2 IGRA TUBE. The limit of detection for the Quan-T-Cell SARS-CoV-2 assay was provided by the manufacturer and is 18.44 IU/mL. Similar cut-off values were used for the Omicron tube, as no official cut-offs were available from the manufacturer at the time of the study. Measurements were categorized as invalid if the negative control was >0.4 IU/mL or the positive control was <0.4 IU/mL after background subtraction. Samples with a detection level above the maximum linear range value were placed with values greater than the largest IFNy value detectable after extrapolation (>8 IU/mL).

### 2.3. Roche Elecsys Anti-SARS-CoV-2 S Measurement

EDTA-plasma samples were used to perform the serologic assays. Samples from 24 subjects were included, 11 BTI and 13 non BTI. Venous samples were taken in 3 mL EDTA plasma tubes (Sarstedt, Nümbrecht, Germany) and mixed by inverting several times. The cell pellet was removed by centrifugation (for 10 min, 2500 rpm) and the plasma was transferred into 2 mL individually barcoded screw cap tubes (Sarstedt, Nümbrecht Germany). We performed the serologic assessment as recommended by the manufacturer. In brief, values are given in Units/mL, the positivity threshold is set to 0.8 as recommended. Values above the linear range specified by the manufacturer (250 U/mL) were diluted as recommended in the manual until the measurements reached linear range again. The final concentration in these cases was calculated using the dilution factor and the measured units. Complete descriptions of the assays used for this analysis have been already published and can be found [16].

### 2.4. Statistical Analysis

The complete dataset was cleaned and locked prior to the conduction of any analyses that were performed in R (version 4.0.5, R Development Core Team, 2021). Overall testing was performed with a Kruskal–Wallis test, while differences between groups were accessed via Mann–Whitney testing. The Graph visualizing the IFNy release (Figure 1) was created in GraphPad Prism Version 6.0.

Continuous variables were plotted as boxplots (Figure 2). 

## 3. Results

In order to determine whether the SARS-CoV-2 IGRA TUBE was able to detect SARS-CoV-2 infection similarly to the SARS-CoV-2 IGRA Omicron tube, the plasma concentration of IFNy was measured. This was performed 6–55 days after the infection, after overnight stimulation of whole blood with the respective antigens. Out of the 26 subjects, 2 had to be excluded from the analysis; 1 subject had an invalid negative control (IFNy > 0.4 IU/mL), while 1 subject had an invalid positive control (IFNy < 0.4 IU/mL). Further, three subjects had a missing Omicron tube resulting in *n* = 21 complete datasets. When it comes to the original SARS-CoV-2 IGRA TUBE, two measurements fell within the borderline range of IFNy detection (0.1–0.2 IU/mL), while SARS-CoV-2 in the rest of the measurements (*n* = 19) was detected (>0.2 IU/mL) (Figure 1). However, when it comes to the SARS-CoV-2 IGRA Omicron tube, 1 measurement was categorized as negative to SARS-CoV-2, 1 measurement was in the borderline range, while 19 measurements were considered SARS-CoV-2 positive (Figure 1). Next, we wanted to determine whether the IFNy production from the Omicron tube might be of higher concentration due to similarity with the strain causing the infection. For this analysis, only subjects that had values in the linear range for both tubes were included (*n* = 16). There was no significant difference in the IFNy production between both tubes (*p* = 0.99) (Figure 1). Furthermore, the ratio of the median difference in IFNy concentration between the Omicron and the IGRA TUBE was 1.04 (IQR: 0.82–1.2).

Next, we compared the influence of Omicron versus Wuhan infection on detection of SARS-CoV-2-specific antibodies using the Roche Elecsys anti-S assay at 2–5 weeks after infection. This assay detects Spike-specific antibodies to the RBD region using a truncated S1 with the original Wuhan antigen sequence. In non-vaccinated (non BTIs) Omicron infected patients (*n* = 12), the assay detected a factor 100-times lower median Spike-specific RBD antibody concentration (*p* < 0.0001) compared to patients at matched time points after diagnosis during May 2020–January 2021 (Figure 2). In contrast, for patients who had been vaccinated against SARS-CoV-2 before Omicron infection (*n* = 10) (BTIs), the Roche Elecsys anti-SARS-CoV-2 anti-S1 assay detected a 400-times higher median specific antibody concentration (*p* < 0.0001) compared to patients referring to the beginning of the pandemic. 

## 4. Discussion

With frequent emergence of new SARS-CoV-2 virus variants, accumulating especially immune evasion mutations in the Spike protein, it is of utmost importance to ensure that current diagnostic tools are up to date and still functional in the detection of the most recent and prevalent virus strains. Furthermore, determining potential shortcomings early is important as the development or update of these diagnostic tools is a lengthy process. In this study, we have decided to analyze the Quan-T-Cell SARS-CoV-2 assay produced by EUROIMMUN. They contain an IGRA stimulator TUBE, containing peptide antigens derived from the Spike protein of the original Wuhan virus strain. By using samples from Omicron infected patients, we compared this original stimulator tube with an updated version of the stimulator tube containing Omicron variant based peptides. 

EUROIMMUN provided both tubes, in order to determine whether the stimulator tube within the kit needs to be adapted to the Omicron variant. Our results show that the original stimulator with the Wuhan antigen was able to detect all the Omicron infected subjects. Furthermore, the concentrations of IFNy produced by the SARS-CoV-2-specific immune response were similar between both stimulator tubes. This can be explained by the T-cells recognizing a variety of epitopes in antigenic regions more conserved within the Spike protein, compared to the more variable RBD region, which contains the majority of Spike-specific antibody escape mutations impacting on RBD recognition. This is consistent with previous results using other techniques, such as intracellular cytokine staining or virus neutralization assays [5]. Nevertheless, this study has some limitations. For example, the relatively small sample size and the different group sizes. This was due to the difficulties to quickly identify and recruit participants—particularly unvaccinated individuals. Future studies should avoid these limitations by including a sample size calculation in the process of the study design and focus on an even recruitment in both groups. In addition, the Roche Elecsys anti-SARS-CoV-2 anti-S1 assay is based on the original Wuhan virus Spike antigen and was not adapted to include Omicron Spike version. 

In conclusion, our results suggest that the current Wuhan antigen based Quant-T-Cell-SARS-CoV-2 kit detects T-cell response to the currently prevalent Omicron variants with similar results to tubes with an Omicron adapted antigen. The data suggest that different test systems show variable performance when used on patients infected with different variants of SARS-CoV-2. An Omicron only infection induces an antibody response that better recognizes Omicron-Spike variants over the Wuhan variants, while Spike-specific T-cell responses are much less affected [5,17,18,19]. As a consequence, commercial diagnostic assays using Wuhan-based antigen for quantifying the Spike-specific antibody response upon Omicron infection may underestimate the variant-specific response magnitude, while assays that detect T-cell responses are much less affected. Using the tests presented here, we can conclude that unvaccinated Omicron first time infected patients for example will show weak and very delayed seroconversions in the Elecsys anti S1 assay, while the IGRA with wild type or Omicron peptides is positive much earlier.

Thus, depending on the clinical question, medical doctors and laboratories need to know the performance characteristics of their test systems in regard to the history of the patient to draw the right conclusions.

Furthermore, the results of our research also suggest that the original Quan-T-Cell kits can still be used in the current Omicron phase of the pandemic. Potential applications of the T-cell analysis could, for example, be to provide immunological insight into the disease dynamics over a longer timeframe, compared to, for example, PCR tests. Furthermore, it can be used to check the immune status and compare breakthrough infections and non-breakthrough infected patients. The assay may also be used to detect asymptomatic infections. Comparing the serologic results of the groups, all Omicron BTI-subjects are far above the positivity threshold. However, infection- and vaccination-naive individuals responded in a much weaker fashion than what was observed in the control group of patients infected with the Wuhan strain. Actually, in the anti-S1 response shown here, after 20–40 days, only 4 out of 12 are above the positivity threshold provided by the manufacturer at the investigated timepoint. Almost all of the patients had seroconverted against Nucleocapsid at the same time point (data not shown). This demonstrates that a detectable serological response was also found, confirming the diagnosis and data obtained with the IGRA assays. However, the measured response in the Roche Anti-S ELECSYS assay, using a truncated S1 as a target structure, are considerably lower compared to the values observed in Wuhan strain infected subjects [20]. Whether this observation is primarily due to differences in binding to the antigen used, or due to less immune- stimulation in the commonly milder Omicron variant infections as compared to the Wuhan strain is unclear and cannot be elucidated within this study. 

## Figures and Tables

**Figure 1 diagnostics-13-01024-f001:**
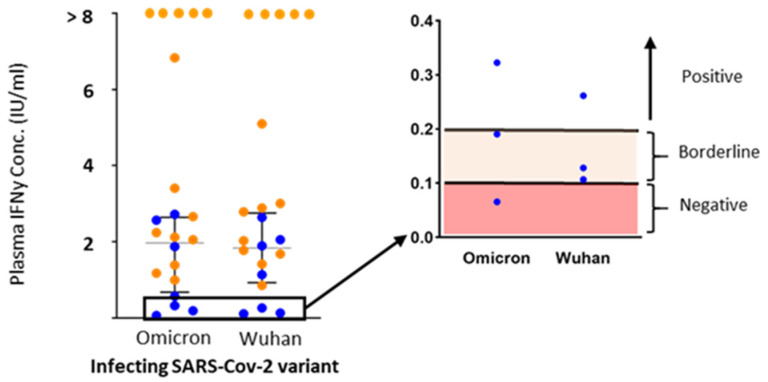
Interferon gamma release upon in vitro restimulation with SARS-CoV-2 Wuhan and Omicron variant antigens. IFNy release was tested in the interferon gamma release assay after overnight stimulation with Omicron Spike variant antigen and Wuhan Spike variant antigen (*x*-axis) in individuals with breakthrough infection (BTI, orange circle, *n* = 14) and non BTI (blue circle, *n* = 7). A zoomed image (black box) shows the subjects that fell within the regions considered as a negative response (<0.1 IU/mL, shaded red) to SARS-CoV-2, and borderline results (0.1–0.2 IU/mL, shaded orange).

**Figure 2 diagnostics-13-01024-f002:**
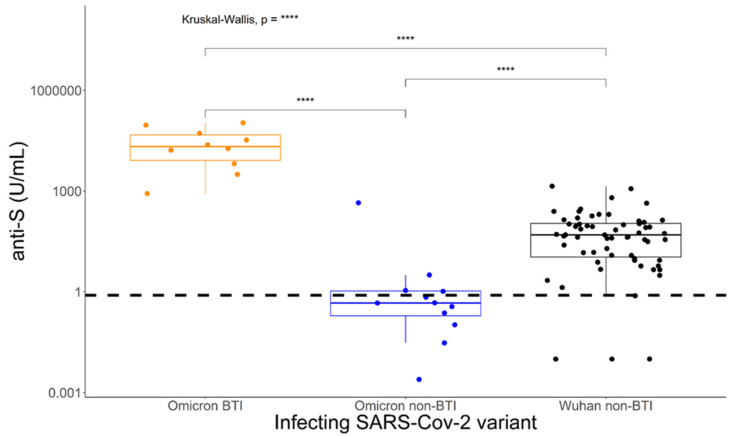
Induction of Wuhan Spike-Receptor Binding Domain-specific antibody concentrations differ between Omicron and Wuhan non-breakthrough infections and Omicron breakthrough infections. Patients were tested at 2–5 weeks after PCR diagnosis of SARS-CoV-2 infection with the Roche Elecsys anti-S assay, which incorporates the receptor binding region of the Wuhan wild type virus. Statistical analyses were performed using the Mann–Whitney U test. **** *p* < 0.0001.

## Data Availability

Further information and data can be requested via mail to the corresponding authors and is not available online due to privacy reasons.

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
