# Peer review of "Impact of Omicron Variant Infection on Assessment of Spike-Specific Immune Responses Using the EUROIMMUN Quan-T-Cell SARS-CoV-2 Assay and Roche Elecsys Anti-SARS-CoV-2-S"

_diagnostics, 2023, doi:10.3390/diagnostics13061024_

Round 1
Reviewer 1 Report
The study design is appropriate to answer the aim. This study added to what is already in the topic. The article is consistent within itself. There are no major flaws associated with this article. Overall strengths of the article are the design and the reliable methods which ensure validated results and conclusions.
Specific comments on weaknesses of the article and what could be improved:
Major points - none
Minor points
1. Please, state the limitations of the study
2. How these results can be translated into the clinical practice? Authors should speculate on recommendations based on these results.
3. There are some minor grammar issues - i.e., page 3, section 2.3., "Samples from 24 subjects could be included" should be "...were included", etc.
4 Could authors consider moving fig. 1 and fig. 2 and related data to the results section. In case they insist this to be covered in the Material and methods, please, give your arguments.
5 Describe more profoundly the inclusion and exclusion criteria for the study subjects.
Author Response
Response to Open Review 1
We would like to thank the reviewer for providing us with feedback for our manuscript and hope to have answered all questions in the following.
- Please, state the limitations of the study
We have included the limitations we identified in our research in the manuscript in the discussion on page 7:
Nevertheless, his study has some limitations. For example, the relatively small sample size and the different group sizes. This was due to the difficulties to quickly identify and recruit participants - particularly unvaccinated individuals. Future studies should avoid these limitations by including a sample size calculation in the process of the study design and focus on an even recruitment in both groups. In addition, the Roche Elecsys assay that was used, is based on the original Wuhan virus Spike antigen and was not adapted to include Omicron Spike version.
- How these results can be translated into the clinical practice? Authors should speculate on recommendations based on these results.
In the following we tried to speculate on potential recommendations based on our results:
The data suggest, that different test systems show variable performance when used on patients infected with different variants of SARS-CoV-2. Using the tests presented here, we can conclude that unvaccinated Omicron first time infected patients for example will show weak and very delayed seroconversions in the ELECSYS anti S1 assay, while the IGRA with wild type or Omicron peptides is positive much earlier. Thus, depending on the clinical question, medical doctors and laboratories need to know the performance characteristics of their test systems in regards to the history of the patient to draw the right conclusions.
Here e.g. it might be superior to use an IGRA in a unvaccinated and Omicron first time infected subject if you want to show immune reactions or confirm diagnosis in retrospect, while the same subject with a breakthrough infection with Omicron after mRNA based vaccination will be positive for very high titres in the ELECSYS Anti S1 and will not require IGRA testing.
We have also tried to include the added value of our research and potential applications in the clinical or scientific practice more detailed in the discussion on page 7:
The data suggest that different test systems show variable performance when used on patients infected with different variants of SARS-CoV-2. An Omicron only infection induces an antibody response that better recognizes Omicron-Spike variants over the Wuhan variants, while Spike-specific T-cell responses are much less affected (5,17–19). As a consequence, commercial diagnostic assays using Wuhan-based antigen for quantifying the Spike-specific antibody response upon Omicron infection may underestimate the variant-specific response magnitude, while assays that detect T-cell responses are much less affected. Using the tests presented here, we can conclude that unvaccinated Omicron first time infected patients for example will show weak and very delayed seroconversions in the elecsys anti S1 assay, while the IGRA with wild type or Omicron peptides is positive much earlier.
Thus, depending on the clinical question, medical doctors and laboratories need to know the performance characteristics of their test systems in regards to the history of the patient to draw the right conclusions.
Furthermore, the results of our research also suggest, that the original Quan-T-cell kits can still be used in the current Omicron phase of the pandemic. Potential applications of the T-cell analysis could for example be to provide immunological insight into the disease dynamics over a longer timeframe, compared to for example PCR tests. Furthermore, it can be used to check the immune status and compare breakthrough infections and non-breakthrough infected patients. The assay may also be used to detect asymptomatic infections.
- There are some minor grammar issues - i.e., page 3, section 2.3., "Samples from 24 subjects could be included" should be "...were included", etc.
We searched for grammatical errors and fixed them
- Could authors consider moving fig. 1 and fig. 2 and related data to the results section. In case they insist this to be covered in the Material and methods, please, give your arguments.
We have moved the figures to the results. They have been submitted at the end of the manuscript but must have been moved in the conversion or download from the submission page.
- Describe more profoundly the inclusion and exclusion criteria for the study subjects.
We tried to describe it more detailed and give further information where to find the criteria the study was based on in the methods section under part 2.1 on page 3:
KoCo19-Immu aims to identify and characterize factors that influence the clinical course and further transmission of SARS-CoV-2 infection. The 37 participants of this KoCo19-sub-study were recruited from December 2021 until the end of March 2022. The general inclusion and exclusion criteria for the KoCoImmu study are based on the ones of the KoCo19 study which have been described in detail previously (15) . Additionally, there were specific criteria used for the purpose of the omicron subgroup. Only outpatient BTI (vaccinated followed by infection, n= 20) and non BTI (first time infected, n=17) were recruited. Potential participants were questioned prior to the visit and only the ones who reported no previous SARS-CoV-2 infection were included. Furthermore, Ffor the purpose of this analysis, only subjects that were confirmed SARS-CoV-2 PCR positive by routine laboratory diagnostics were considered. and had all necessary samples takn were used. Subjects were either reported by the health au-thorities as confirmed omicron cases or had a confirmation of omicron infection by the initial PCR test or a high likelihood of an omicron infection, which was indicated by testing positive for a mutation marker. No PCR confirmation or a different virus vari-ant were exclusion criteria. Recruitment into the study followed 3 weeks after the SARS-CoV-2 PCR positive diagnosis. Missing necessary samples were classified as ex-clusion criteria. Samples from acutely SARS-CoV-2 infected subjects recruited during the early phase of the pandemic (May 2020-January 2021) were matched according to the time since diagnosis for serological comparisons with the Omicron cases.

Reviewer 2 Report
1. The mutation of SARS-CoV-2 spike proteins is easily and frequent; thus, the nucleocapsid and the envelope protein are better targets than spike proteins for viral detection. Especially, the emergency of Omicron viral variants is more rapid than the original Wuhan or Delta strains because of the mutation of spike proteins. In this manuscript, the authors want to report the impact of Omicron variant infection on assessment of spike-specific immune responses using the Quan-T-Cell SARS-CoV-2 assay and the Roche Elecsys anti-SARS-CoV-2 S measurement. However, both the methods are based on the Omicron spike protein. Please show that how to maintain the accuracy, specificity and precision for viral detection using these two assay if there are more new Omicron variants emergency.
2. Please indicate how to get the limit of detection (LOD) of the Quan-T-Cell SARS-CoV-2 assay and provide their results in the “Material and Methods” and “Result” section, respectively.
3. Please indicate how to get the limit of quantitation (LOQ) of the Roche Elecsys anti-SARS-CoV-2 S measurement and provide their results in the “Material and Methods” and “Result” section, respectively.
4. Please discuss the novelty, significance and contribution to scientific community to assess the using the impact of Omicron variant infection using the Quan-T-Cell SARS-CoV-2 assay and the Roche Elecsys anti-SARS-CoV-2 S measurement, compared to the traditional methods.
Author Response
Response to Open Review 2
We would like to thank the reviewer for providing us with feedback for our manuscript and hope to have answered all questions in the following.
- The mutation of SARS-CoV-2 spike proteins is easily and frequent; thus, the nucleocapsid and the envelope protein are better targets than spike proteins for viral detection. Especially, the emergency of Omicron viral variants is more rapid than the original Wuhan or Delta strains because of the mutation of spike proteins. In this manuscript, the authors want to report the impact of Omicron variant infection on assessment of spike-specific immune responses using the Quan-T-Cell SARS-CoV-2 assay and the Roche Elecsys anti-SARS-CoV-2 S measurement. However, both the methods are based on the Omicron spike protein. Please show that how to maintain the accuracy, specificity and precision for viral detection using these two assay if there are more new Omicron variants emergency.
The ELECSYS anti-S1 assay is based on the Wuhan (wild type) strain, not the Omicron variant as the reviewer suggests. Similarly, only one of the tubes used in the IGRA is based in the Omicron variant, the others are controls or Wuhan strain based. Actually, a head to head testing was performed in regards to the IGRA with an Omicron and a Wuhan tube. In ELECSYS, this was not performed as an Omicron variant version of the test is currently not available (for spike or nucleocapsid). It can be seen that the IGRA tubes with both different peptide pools performed similarly. For the serology the seroconversion against S1 (Wuhan) was delayed as compared to infections with Delta, Alpha or Wuhan variants. The exact reason is unclear but might be due to less reactivity against the antigen and less immuno stiumlation due to mild disease, or a mix of both, this is discussed in the discussion section.
- Please indicate how to get the limit of detection (LOD) of the Quan-T-Cell SARS-CoV-2 assay and provide their results in the “Material and Methods” and “Result” section, respectively.
The value was included in the manuscript in the Methods, section 2.2 on page 3.
The limit of detection for the Quan T-Cell SARS-CoV-2 assay was provided by the manufacturer and is: 18,44 mIE/ml. The Limit of Quantitation for the test is also given by EUROIMMUNE and is: (LoQ): 31,07 mIE
- Please indicate how to get the limit of quantitation (LOQ) of the Roche Elecsys anti-SARS-CoV-2 S measurement and provide their results in the “Material and Methods” and “Result” section, respectively.
The LOQ is 0.4-250 U/ml according to the manufacturer, the cutoff is 0.8U/ml. In our hands we have good results between 0.1 and about 400 IU/ml (own linearity data, not for the purpose of this study). As the data used in the publication is based on the manufacturers ranges, and it is licensed as a quantitative test, no further information is needed to be presented, it can be taken from the package insert of the test system for any inclined user of the system. Values above 250 U/ml were diluted until the linear range was reached and calculated back to give the correct reading in U/ml to obtain reliable data also for higher titres.
We have added the following paragraph to the methods section 2.3 on page 4:
Values above the linear range specified by the manufacturer (250U/ml) were diluted as recommended in the manual until the measurements reached linear range again. The final concentration in these cases was calculated using the dilution factor and the measured units.
- Please discuss the novelty, significance and contribution to scientific community to assess the using the impact of Omicron variant infection using the Quan-T-Cell SARS-CoV-2 assay and the Roche Elecsys anti-SARS-CoV-2 S measurement, compared to the traditional methods.
We tried to show the added value of our research and added it in the discussion on page 7 as follows:
The data suggest that different test systems show variable performance when used on patients infected with different variants of SARS-CoV-2. An Omicron only infection induces an antibody response that better recognizes Omicron-Spike variants over the Wuhan variants, while Spike-specific T-cell responses are much less affected (5,17–19). As a consequence, commercial diagnostic assays using Wuhan-based antigen for quantifying the Spike-specific antibody response upon Omicron infection may underestimate the variant-specific response magnitude, while assays that detect T-cell responses are much less affected. Using the tests presented here, we can conclude that unvaccinated Omicron first time infected patients for example will show weak and very delayed seroconversions in the elecsys anti S1 assay, while the IGRA with wild type or Omicron peptides is positive much earlier.
Thus, depending on the clinical question, medical doctors and laboratories need to know the performance characteristics of their test systems in regards to the history of the patient to draw the right conclusions. Furthermore, the results of our research also suggest, that the original Quan-T-cell kits can still be used in the current Omicron phase of the pandemic. Potential applications of the T-cell analysis could for example be to provide immunological insight into the disease dynamics over a longer timeframe, compared to for example PCR tests. Furthermore, it can be used to check the immune status and compare breakthrough infections and non-breakthrough infected patients. The assay may also be used to detect asymptomatic infections.

Round 2
Reviewer 2 Report
No